# *"I got a bunch of weed to help me through the withdrawals"*: Naturalistic cannabis use reported in online opioid and opioid recovery community discussion forums

**Meredith C. Meacham**[1]*, **Alicia L. Nobles**[2], **D. Andrew Tompkins**[1], **Johannes Thrul**[3,4,5]

**1** Department of Psychiatry and Behavioral Sciences, University of California San Francisco, San Francisco, CA, United States of America, **2** Department of Medicine, Division of Infectious Diseases and Global Public Health, University of California San Diego, San Diego, CA, United States of America, **3** Department of Mental Health, Johns Hopkins Bloomberg School of Public Health, Baltimore, MD, United States of America, **4** Sidney Kimmel Comprehensive Cancer Center at Johns Hopkins, Baltimore, MD, United States of America, **5** Centre for Alcohol Policy Research, La Trobe University, Melbourne, Australia

* Meredith.meacham@ucsf.edu

**Data Availability Statement:** The data underlying the results presented in the study are available

## Abstract

A growing body of research has reported on the potential opioid-sparing effects of cannabis and cannabinoids, but less is known about specific mechanisms. The present research examines cannabis-related posts in two large online communities on the Reddit platform ("subreddits") to compare mentions of naturalistic cannabis use by persons self-identifying as actively using opioids versus persons in recovery. We extracted all posts mentioning cannabis-related keywords (e.g., "weed", "cannabis", "marijuana") from December 2015 through August 2019 from an opioid use subreddit and an opioid recovery subreddit. To investigate how cannabis is discussed at-scale, we identified and compared the most frequent phrases in cannabis-related posts in each subreddit using term-frequency-inverse document frequency (TF-IDF) weighting. To contextualize these findings, we also conducted a qualitative content analysis of 200 random posts (100 from each subreddit). Cannabis-related posts were about twice as prevalent in the recovery subreddit (n = 908; 5.4% of 16,791 posts) than in the active opioid use subreddit (n = 4,224; 2.6% of 159,994 posts, *p* < .001). The most frequent phrases from the recovery subreddit referred to time without using opioids and the possibility of using cannabis as a "treatment." The most frequent phrases from the opioid subreddit referred to concurrent use of cannabis and opioids. The most common motivations for using cannabis were to manage opioid withdrawal symptoms in the recovery subreddit, often in conjunction with anti-anxiety and GI-distress "comfort meds," and to enhance the "high" when used in combination with opioids in the opioid subreddit. Despite limitations in generalizability from pseudonymous online posts, this examination of reports of naturalistic cannabis use in relation to opioid use identified withdrawal symptom management as a common motivation. Future research is warranted with more structured assessments that examines the role of cannabis and cannabinoids in addressing both somatic and affective symptoms of opioid withdrawal.

from the Pushshift Reddit Dataset: https://ojs.aaai.
org//index.php/ICWSM/article/view/7347.

**Funding:** This research was supported by U.S.
National Institute of Drug Abuse grants
K01DA046697 (MCM), K25DA049944 (ALN),
R21DA047520 (DAT), and R21 DA048175 (JT).
URL: https://www.drugabuse.gov/. The funders
had no role in study design, data collection and
analysis, decision to publish, or preparation of the
manuscript.

**Competing interests:** I have read the journal's
policy and the authors of this manuscript have the
following competing interests: In the last 3 years,
DAT has received consulting fees from Alkermes,
Inc. and Clinilabs. The other authors have declared
that no competing interests exist. This does not
alter our adherence to PLOS ONE policies on
sharing data and materials.

## Introduction

The opioid epidemic in North America is characterized by dramatic increases in opioid-related overdoses [1] and unmet treatment need for opioid use disorder (OUD) [2]. As Canada and most U.S. states enact decriminalization and legalization of cannabis for medical and recreational adult use, there is considerable interest in determining whether cannabis and cannabinoid products are viable alternatives to opioids for pain management or a potential adjunctive therapy for OUD. Recent interest is spurred in part by several studies that found relationships between cannabis legalization at the state level and reductions in opioid-related deaths [3–8], though the ecological association between state-level medical cannabis legislation and reduced opioid overdose mortality reported previously [3] was not replicated in a study examining additional years [9]. Additionally, several studies have found that medical cannabis patients report pain management as one of the most common reasons for cannabis use [10–12] and that cannabis use was associated with lower opioid use among people who use illicit drugs [13, 14]. However, there are also ongoing concerns about cannabis use increasing the risk for non-medical opioid use and opioid and cannabis use disorders, especially among adolescents and young adults [15–18], and that cannabis use may lead to return to opioid use by people with a history of OUD [19–22].

The National Academies of Science, Engineering, and Medicine (NASEM) 2017 report on the Health Effects of Cannabis and Cannabinoids found some evidence that cannabinoids may be an effective treatment for some types of chronic pain in adults, but no evidence to support or refute the use of cannabinoids for treating addiction [23]. One reason for the limited scientific evidence is that cannabis is a Schedule I drug in the United States with "no currently accepted medical use," limiting funding and approval to study real world cannabis as a potential therapy in randomized controlled trials [24].

In the meantime, a wide range of experiences with substances are shared on social media. These unsolicited reports have become an alternative information source to surveys, administrative, and clinical data regarding the use of cannabis, opioids, and other substances in natural settings [25]. One place where people share experiences with illicit substance use is Reddit, a social media and news aggregation platform that allows users to interact through topic-specific forums called subreddits, instead of around personal profiles or hashtags as with Twitter or Facebook. In addition to the subreddit organization, users contribute through pseudonyms, which may lead to greater tendencies to disclose hidden or stigmatized behaviors than on other identity-linked social media platforms. The Pew Research Center reports that in 2019 over 1 in 10 U.S. adults used Reddit [26]. In 2020, Reddit was the 7th most visited website in the United States [27], with most of the global traffic to Reddit coming from the United States and Canada.

Both qualitative and computational approaches have been used to examine discussions about cannabis and opioid use on Reddit. Content analyses have reported banter, disclosure, instruction, and advice as common types of posts and comments in cannabis and opioid use subreddits [28], and have identified posters disclosing DSM-5 criteria for cannabis use disorder [29] and opioid use disorder [30] in cessation and recovery-oriented subreddits. A thematic analysis of posts to popular opioid-related subreddits during the early stages of the COVID-19 pandemic found evidence of robust mutual aid and social support [31]. In a computational text analysis across 1.4 million Reddit posts in 63 subreddits, previous work identified common terms used to describe alternative treatments for opioid use recovery, the most common of which was kratom [32]. Through a word embedding analysis, that study found that cannabis had the highest cosine similarity with kratom (indicating that the word "cannabis" was the most contextually similar word to "kratom" in these opioid recovery

posts). In another analysis of 2.3 million posts to a large cannabis subreddit, emerging forms of cannabis (vaping, edibles, concentrates, dabbing) were increasingly mentioned from 2011 to 2016 [33]. However, no study to our knowledge has focused specifically on the intersection between cannabis and opioid content on Reddit.

### Objective

In this study, we apply computational and qualitative research methods to assess and compare between an active opioid use subreddit and an opioid recovery subreddit: 1) the proportion of posts that mention cannabis, 2) the most common words and phrases in posts that mention cannabis, and 3) motivations for cannabis use in relation to opioid use as described in cannabis-related posts.

## Materials and methods

### Ethical considerations

Both subreddits from which sampled posts were drawn are publicly accessible and findings are presented in aggregate. This research was categorized as human subject exempt category 4 by the University of California San Francisco Institutional Review Board. Usernames were not examined. Sample quotations are composites or lightly reworded to reduce the risk of identification.

### Data collection

All posts (title and content) and associated metadata posted from December 2015 to August 2019 (45 months) on an opioid use subreddit and an opioid recovery subreddit were downloaded from the Pushshift Reddit database [34] via Google Big Query. Data were analyzed in R Studio version 1.2.5019. Duplicate posts were removed (1.5% of posts), which were identified with the "duplicated" base R function applied to post text.

### Proportion of cannabis-related posts in each subreddit

Posts mentioning cannabis were selected using a keyword search strategy that identified whether the post contained at least one of eight cannabis-related terms (weed, cannabis, marijuana, pot, reefer, ganja, thc, and cbd). A sample of posts were read individually to see if these posts were indeed cannabis related. To avoid capturing false positives detected through this sample reading (e.g., "spot," "potency," and "healthcare"), white space was added around "pot" and "thc." In later readings of individual posts, the word "weed" picked up some additional false positives (e.g., "pulling weeds"), but also some typos (e.g., "weeds" instead of "weed is").

The proportion of cannabis-related posts in each subreddit was compared using a chi-square test. The median number of words per a cannabis-related post was compared between subreddits using a Wilcoxon rank sum test. Given the increase in the number of U.S. states with cannabis legalization during the study period (2015–2019), the potential change in the frequency of cannabis-related posts over time was examined using linear regressions with proportion of cannabis-related posts per month (relative to all posts) as a function of time in months.

### Most frequent and distinctive words and phrases in cannabis-related posts

First, the most frequent words and phrases in the cannabis-related posts for each subreddit were determined using term frequency-inverse document frequency (tf-idf). Second, the most distinctive words and phrases contained in the cannabis-related posts for each subreddit were identified using likelihood ratio for a keyness analysis [35].

Punctuation, numbers, urls, symbols, html strings, and stop words (e.g., prepositions) were removed from the cannabis-related posts. Then the post was tokenized into unigrams (singular words like "opioid"), bigrams (two-word phrases like "opioid use"), and trigrams (three-word phrases like "opioid use disorder"). After this pre-processing, weighted frequency of each n-gram was ranked using TF-IDF, which balances the occurrence of an n-gram in a post relative to its frequency in the entire corpus of posts. TF-IDF avoids the potential bias of using just term frequency (i.e., raw counts of n-grams). The top 10 unigrams, bigrams, and trigrams, identified by the highest TF-IDF, were identified for each subreddit.

Next, the textstat_keyness function in the quanteda [36] package in R was used to identify the most distinctive unigrams and bigrams of each subreddit. This function calculates a "keyness" score for each n-gram indicating its uniqueness in a "target" corpus (i.e., the cannabis-related posts in the recovery subreddit) compared to a reference corpus (i.e., the cannabis-related posts in the opioid use subreddit), specified as a likelihood ratio ($G^2$) [37, 38] with William's continuity correction. Keyness scores were sorted by $G^2$, and the unigrams and bigrams with the 15 highest and lowest $G^2$ values were identified.

### Motivations for cannabis use

To further contextualize these automated text analysis results, we conducted a qualitative analysis of 200 random cannabis posts (100 from each subreddit). A codebook was initiated by MM with open coding of random 20 posts per subreddit with the goal of identifying (1) disclosure of cannabis and/or opioid use and (2) motivations for using cannabis (or not) in relation to using opioids or the reduction or cessation of opioid use. This initial codebook was shared and discussed with other investigators (AN, JT). A refined codebook was then applied independently to 50 random posts (25 from each subreddit) by three investigators (MM, AN, JT) with substance use-specific expertise in public health, information science, and psychology, respectively. Investigators were blinded to the original subreddit source. The three sets of applied codes were compared and discussed over email and in video conference meetings. Discordant codes were flagged and discussed, and the codebook was refined one more time and applied again to the same 50 posts. Inter-rater agreement was calculated (.82–1.00), and discordant codes were discussed until consensus was reached.

A final codebook was applied by three investigators (MM, AN, JT) to an additional 50 posts (150 single coded posts + 50 triple coded posts), split evenly by subreddit and with investigators blinded to subreddit source. The final codebook included prompts, definitions, subcodes, and quotation examples of 10 codes, summarized in Table 4. Codes included noting false positives, disclosure and timing of opioid or cannabis use, and whether CBD (cannabidiol) was the form of cannabis used, given growing interest in CBD [39] and the legalization of hemp-derived CBD in the United States in early 2018. Motivations for cannabis use in relation to opioid use included using cannabis to manage opioid withdrawal symptoms, at the same time as an opioid (co-use/ polysubstance use), or for pain management, as well as questions about cannabis use in longer term desistence of opioid use or self-identified recovery. Interactions with healthcare providers in general and regarding cannabis were also noted.

Percentage of times the codes were applied were tabulated and compared between subreddits. Posts with similar codes were read together to confirm coding consistency, explore intra code variability, and examine contextual factors.

## Results

### Proportion of cannabis-related posts

Over 45 months from late 2015 to mid 2019, there were 16,791 total posts to the recovery subreddit and 159,994 posts to the opioid subreddit. Cannabis-related posts were about twice as

**Table 1. Comparison of posts containing cannabis terms between active opioid use and opioid recovery subreddits (late 2015-mid 2019).**

|  | Recovery Subreddit | | Opioid Subreddit | |
|---|---|---|---|---|
| Total Posts (N) | 16,791 | | 159,994 | |
| Cannabis Posts (N, % of Total) | 908 | 5.4% | 4,224 | 2.6% |
| Total Wordcount (Median, IQR) | 101 | (32, 224) | 24 | (8, 88) |
| Cannabis Post Wordcount (Median, IQR) | 279 | (151, 435) | 173 | (80, 347) |
| Change in proportion of posts mentioning cannabis per month (Slope, *p*-value) | 0.014% | 0.29 | -0.013% | 0.0051 |

IQR: Inter-quartile range.

prevalent in the recovery subreddit (n = 908, 5.4%) than in the opioid subreddit (n = 4224, 2.6%) (*p* < .001). Total number of posts, proportion of posts mentioning cannabis, and word count distributions are presented in Table 1. The median wordcount for cannabis-related posts in the recovery subreddit was 279 words, which was significantly longer than the median wordcount of 173 words for cannabis-related posts in the opioid subreddit (*p* < .001). There was no significant change over these 45 months in proportion of cannabis-related posts in the recovery subreddit. However, there was a slight decrease in proportion of cannabis related posts in the opioid subreddit. When 17 false positive posts containing the string "weeds" were removed from counts of cannabis posts, analyses of proportions were no different.

## Most frequent and distinctive words and phrases in cannabis-related posts

The top 10 most frequent unigrams, bigrams, and trigrams in cannabis-related posts from each subreddit are presented in Table 2. Of note, the most frequent phrases in the recovery subreddit included mentions of cannabis as a "treatment" or "substitution." References to "recovery," "substitution," and "treat" and words indicating amount of time ("days", "years") were also frequent. "CBD" is mentioned along with "kratom" and "benefits" and longer phrases referred to using "marijuana [to] treat addiction", "success stories [with] marijuana," and "planning [to] taper Subutex." This is reflective of posts commonly discussing time in treatment and recovery, often reporting or questioning using cannabis as a "treatment."

The most frequent words and phrases in the opioid subreddit included "smoking weed" and combinations of different substances (opioids, cannabis, alcohol, stimulants). References of "CBD" were in n-grams with hydrocodone and oxycodone. Specific anti-anxiety pills ("Xanax", "Valium") and opioid pills ("oxy", "Tramadol", "Dilaudid", "hydro") and

**Table 2. Top 10 most frequent n-grams in cannabis-related posts.**

|  | Top 10 most frequent terms (tf-idf weighting) |
|---|---|
| Recovery unigrams | marijuana, cbd, clean, opiate, recovery, cannabis, kratom, day, days, weed |
| Recovery bigrams | day marijuana, weed recovery, zoloft pot, health benefits, medical marijuana, marijuana treat, opiate substitution, substitution thc, benefits cannabis, weed social |
| Recovery trigrams | opiate substitution thc, health benefits cannabis, weed social drinking, kratom cbd recovery, medical marijuana treat, marijuana treat addiction, health benefits cbd, benefits cbd oil, facts opiate crisis, opiate crisis cbd |
| Opioid unigrams | opiates, marijuana, cbd, heroin, weed, like, smoke, oxy, just, get |
| Opioid bigrams | smoke weed, smoking weed, weed opiates, first time, medical marijuana, hydro weed, weed hydro, feel like, anyone else, right now |
| Opioid trigrams | pill porn weed, coke h weed, combining tramadol weed, going weed withdrawals, weed valium amazing, xanax ambien weed, growing cannabis mistake, bars weed hash, h alc weed, 10mg hydro weed |

**Table 3. Top 15 most unique unigrams and bigrams in cannabis-related posts.**

| | Top 15 most unique terms (unweighted) |
|---|---|
| Recovery unigrams | recovery, clean, day, days, help, years, withdrawal, support, paws, life, relapse, using, months, feel, na |
| Recovery bigrams | days clean, feel like, cold turkey, w d [withdrawals], every day, per day, cbd oil, months clean, first days, enough enough, physical symptoms, favorite lyrics, hard time, stay strong, comfort meds |
| Opioid unigrams | bag, guys, guy, black, white, u, lol, fent, fuck, tolerance, dude, weed, shit, opium, dope |
| Opioid bigrams | front door, cold cop, gas station, r drugs, high like, get addicted, shit like, opiate tolerance, anyone else, living room, opium poppy, stay safe, happy nods, drug test, shit post |

formulations of heroin ("h", "tar") are also mentioned. Frequent phrases also noted interactions between opioids and cannabis ("weed potentiate opiates", "combining tramadol weed"). This is reflective of posts commonly discussing active polysubstance use and inquiring about the potential for these substances to interact.

In general, cannabis-related posts in the recovery subreddit contained terms regarding the use of cannabis as a potential treatment, while cannabis-related posts in the opioid subreddit contained terms referring to concurrent use of cannabis and opioids.

Table 3 presents the top 15 most unique unigrams and bigrams relative to the other subreddit. In contrast to the most frequent terms in Table 2, the most unique terms in Table 3 contain fewer cannabis-related terms. Unique recovery subreddit terms in cannabis posts mention time "clean," withdrawal, paws [post-acute withdrawal syndrome], and relapse, as well as challenges with recovery ("physical symptoms," "hard time") and messaging to readers to "stay strong." Unique opioid subreddit terms in cannabis posts mention people ("guy", "dude"), profanity, "tolerance", and references to amounts or quality of drugs or effects ("bag", "high like"). Messaging to other readers are reminders to "stay safe" and have "happy nods."

## Motivations for cannabis use

The most common motivations for using cannabis were to manage opioid withdrawal symptoms in the recovery subreddit and to enhance the high when used in combination with opioids in the opioid subreddit. People described using cannabis products, sometimes as CBD oil, as one of many "comfort meds" to manage withdrawal associated anxiety, insomnia, malaise, and pain both during acute withdrawal and longer-term post-acute withdrawal syndrome (PAWS) [40, 41]. Most people, but not everyone, who described using cannabis for withdrawal symptom management described it as being helpful, but often in the context of a list of other medications and strategies. People also questioned the effectiveness of using cannabis in recovery and seemed to prefer interacting with healthcare providers with whom they could discuss their cannabis use. A summary of how often each disclosure or motivation code was applied is presented in Table 4, with Venn diagrams showing overlaps in motivations disclosed in Fig 1. Sample quotations are presented in Table 5.

## Disclosure—Opioid use

Personal opioid use was disclosed in the majority of sampled posts and at a similar proportion in the recovery subreddit compared to the opioid subreddit (88/100 in recovery subreddit, 78/100 in opioid subreddit; *p = .06*). Posts that did not include personal disclosure described use by a friend or family member or commented on depiction of opioid use in the media.

**Table 4. Disclosures of and motivations for cannabis use in relation to opioid use in each subreddit sample (N = 200 posts).**

| | Code Definition | Recovery Subreddit (N = 100) | Opioid Subreddit (N = 100) |
|---|---|---|---|
| **False Positive** | Post is not about cannabis (e.g., "pulling weeds") | 2 | 0 |
| **Opioid use disclosure** | | | |
| No / Not personal / NA | Refers to someone else's use or media depiction | 11 | 22 |
| Yes | Discloses personal use | 89 | 78 |
| Past | Reflection on initial use or use several years ago | 6 | 9 |
| Current | Recent use, just used, or short-term plans to use | 9 | 40 |
| Reducing | In withdrawal, tapering use, or abstinent | 74 | 29 |
| **Cannabis use disclosure** | | | |
| No / Not personal / NA | Refers to someone else's use or to cannabis legalization news | 23 | 32 |
| Yes | Discloses personal use | 77 | 68 |
| Past | Reflection on initial use or use several years ago | 14 | 14 |
| Current | Recent use, just used, or short-term plans to use | 52 | 49 |
| Reducing | Cutting back or intentionally abstinent | 11 | 5 |
| **CBD use disclosure** | Specifically mentions CBD use | 7 | 2 |
| **Motivations** | | | |
| Withdrawal management | Cannabis is used to manage opioid withdrawal symptoms (e.g., anxiety, nausea, aches, malaise) | 43 | 12 |
| Helpful on its own | States that cannabis is helpful | 9 | 2 |
| Helpful with other strategies | Describes several strategies, including cannabis | 22 | 10 |
| Unclear | Unclear how helpful cannabis is for withdrawal | 10 | 0 |
| Not helpful | States that cannabis is not helpful | 2 | 0 |
| Questioning compatibility | Questions about effectiveness or social acceptability of cannabis use during opioid recovery | 12 | 3 |
| Polysubstance use | Use of cannabis and an opioid at the same time to achieve desired high | 3 | 18 |
| Pain management | Cannabis use explicitly for pain management | 6 | 5 |
| Helpful with other strategies | Describes several strategies, including cannabis | 3 | 2 |
| Unclear | Unclear how helpful cannabis is for pain | 3 | 3 |
| **Healthcare provider interaction** | Poster refers to interaction with a healthcare provider or system | 26 | 14 |
| Related to cannabis use | Refers to a provider or program's attitudes towards cannabis | 7 | 3 |

Within the recovery subreddit the most common disclosure (74/88, 84%) was about efforts to reduce opioid use ("*10 days off today . . . it's a miracle*"). In contrast, within the opioid subreddit the most common disclosure (40/78, 51%) was about recent or planned use ("*Just did a speedball woohoo. . .. Talk to me!*") and there were fewer disclosures (29/78, 37%) of efforts to reduce opioid use ("*The time has come for me to quit heroin*").

## Disclosure—Cannabis use

Disclosure of any cannabis use was similar across both subreddits: 77/100 posts in the recovery subreddit and 68/100 posts in the opioid subreddit disclosed personal cannabis use ($p = .68$). In both subreddits the most common type of disclosure was recent or planned use (52/77, (68%) in the recovery subreddit, 49/68 (72%) in the opioid subreddit). Disclosures of past cannabis use (14 in each subreddit) often referred to initial life experiences with cannabis ("*Began smoking weed in high school*"). In both subreddits (14/77 (18%) recovery, 5/68 (7%) opioid) people mentioned cutting back or intentionally avoiding cannabis use as part of efforts to reduce the use of all substances ("*I'm a week clean of all substances including weed*").

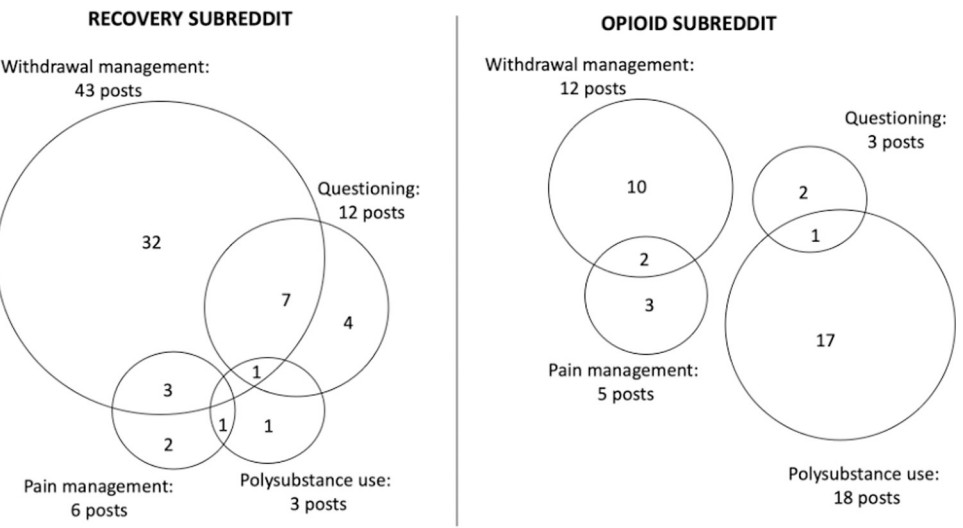

**Fig 1. Overlaps in motivations for cannabis use in relation to opioid use.**

## Motivations—Withdrawal symptom management

The most common motivation for cannabis use was withdrawal symptom management. Almost half the posts (43/100) in the recovery subreddit and a small number of posts (12/100) in the opioid subreddit mentioned using cannabis to manage symptoms of withdrawal like anxiety, nausea, body aches, insomnia, and general malaise.

**Table 5. Sample quotations illustrating motivations for cannabis use in relation to opioids.**

| Motivations | Sample Quotations |
|---|---|
| Withdrawal symptom management | *"Marijuana is AMAZINGLY helpful. I'm going to ramble because it is a nice distraction. I hate marijuana usually. . ..I had some pretty bad WDs and I feel better than ever. . . because of weed." (recovery subreddit)* |
| | *"2 days clean. I have no one else to tell. . . . I'm taking a few hits of pot every 4 hours to ease the symptoms . . . it's been making it tolerable." (recovery subreddit)* |
| Questioning compatibility | *"As far as marijuana goes, I haven't decided if I want to start again. Any feedback from people who have stopped marijuana and then started again would be appreciated. Any regrets starting again? Did it affect your recovery negatively?" (recovery subreddit)* |
| | *"Opiates are usually my first priority, but I'm interested in weed now. My options are either maintenance therapy without weed or to trying to medicate with weed. . . .. Should I keep pursuing maintenance? Or really try to get clean and smoke weed instead? Maybe it's just that now that I'm on subs I see maintenance does not mean I am high all the time and I still want to keep doing that somehow." (recovery subreddit)* |
| Polysubstance use | *"I'm having a very pleasant night sitting here, vegging out online—I am currently stable on my Suboxone dose, with a nice shot of crystal and some good weed. . . ." (opioid subreddit)* |
| | *"Tonight I smoked up a joint of heady weed with hash in it, then smoked like 500mg heroin 50mg at a time." (opioid subreddit)* |
| Pain management | *"Hi chronic pain sufferers—how effective is weed for your pain? . . . does it work for you at all? I find that it is just a distraction for me when I get random back pain." (opioid subreddit)* |
| | *"I came off methadone 9 days ago, and the withdrawal symptoms have sucked, sneezing, watery eyes, restless legs. You know, all that fun stuff. I've been using medicinal cannabis to control the pain, but I'm getting really fed up. Has anyone else done this, and if so, about how long does methadone withdrawal usually last?" (recovery subreddit)* |

Direct quotations are lightly edited to reduce chances of re-identification.

Sometimes people in both subreddits who were anticipating going into withdrawal shared their plan to use cannabis when they ran out of an opioid or as they were intentionally trying to quit without medical assistance: "*The plan after today . . . is to use weed and alcohol to help with the withdrawal symptoms.*"

Other people were seeking immediate advice and support because they were experiencing acute withdrawal symptoms. Common motivations for cannabis use were non-specific, including to mitigate general withdrawal discomfort and distress (*"I'm taking a few hits of pot every 4 hours . . . it's been making it tolerable"*) and symptom-specific, including to address insomnia, nausea, low appetite, anxiety, and restless leg syndrome ("*Restless leg is my worst enemy . . .. Impossible to get any sleep . . . I plan to use benadryl and thc for sleep"*).

Notably, cannabis use was often mentioned in conjunction with what people referred to as "comfort meds" like benzodiazepines and other medications for anxiety and Imodium/lopera-mide for diarrhea. People also mentioned using gabapentin, anti-emetics, anti-hypertensives, sleep medications, and over the counter (OTC) pain medications (ibuprofen, acetaminophen), as well as alcohol and herbs like kratom, kava, and valerian root. Occasionally people mentioned trying meditation, vitamins, easy to digest foods, water, exercise, and social support: "*I detoxed totally cold turkey with nothing but vitamins, exercise, the Good Lord's love and guidance, and a ton of weed.*" While the majority of posts describing cannabis use for withdrawal symptom management were positive towards cannabis (42/55), it was not always clear whether the perceived helpfulness was due to cannabis alone or to other strategies.

A few people, primarily in the recovery subreddit, described not using opioids for over a month but still feeling withdrawal symptoms, often referred to as post-acute withdrawal syndrome (PAWS): "*How do you function while going through paws? . . .. I get to 45 days and still feel terrible and then I relapse. Weed is the only thing that helps"* Some people who had not used in several weeks to months also described how cannabis did not help *("Day 75 . . . weed doesn't even help much [with low appetite and nausea]"*) or even made them feel worse: "*What to do with your life after drugs? . . .. Recently I've turned to weed to help with the anxiety and boredom but that is making me feel like more of a piece of crap.*"

Nine posts mentioned CBD specifically, primarily as being helpful for withdrawal symptoms ("*WD anticipation . . . I have a few Xanax and Valium and some cbd weed to ease the pain*"). Most mentions of CBD were in the recovery subreddit sample (7/100) and many referred to "CBD oil" as the method of use ("*Detoxing off heroin . . . I also have just started taking CBD oil about two weeks ago which has been helping my anxiety a lot")*. Three of these posts described using CBD alone with THC-containing cannabis.

### Motivations—Questioning compatibility with recovery

About 1 in 10 posts in the recovery subreddit sample (12/100) and very few (3/100) in the opioid subreddit sample posed questions to the subreddit members soliciting their opinions and experiences on the effectiveness and social acceptability norms surrounding using cannabis in recovery: *Marijuana and recovery from opiates? What's your idea of "clean"? I know it's a sensitive subject for some people but I'm just looking for honesty and maybe some personal experience stuff*. In contrast, one person noted how using cannabis while in recovery from opiates may have been a trigger for returning to use: "*Smoking weed is all good until you get a case of the fuck-it's and go on a run!*"

### Motivations—Polysubstance use

About 1 in 5 posts in the opioid subreddit (18/100) but very few posts in the recovery subreddit (3/100) described using cannabis and an opioid at the same time. Sometimes people shared

descriptions ("*55 Percocet 7.5mg and 50 100mg thc indica*") or images of substances they were looking forward to using *("Gonna be a fun weekend")* with the intention of enhancing their high and feeling pleasure. Others reported their recent habits or experiences combining opioids and cannabis and how they felt afterwards.

### Motivations—Pain management

A similar proportion of posts in each subreddit (6/100 of recovery subreddit posts, 5/100 of opioid subreddit posts) mentioned using cannabis to control pain due to a chronic pain condition or for painful withdrawal symptoms (e.g., body aches), with mixed opinions on its effectiveness in alleviating pain vs. serving as a mental distraction. As with withdrawal symptom management, cannabis was one several strategies people used to manage pain. Other people mentioned injury or chronic pain as being related to the first time they used opioid pills (but not in relation to cannabis use).

### Healthcare provider interactions regarding cannabis

Health providers or programs were referred to in both subreddits (26/100 in recovery subreddit, 14/100 in opioid subreddit), as people shared past experiences with rehab programs or specific providers or noted an upcoming medical appointment. However, cannabis was mentioned in only a quarter of these interactions (10/40), mostly in the recovery subreddit sample (7/10). Some people related their provider's stance on their cannabis use, with preference for providers that allow them to use cannabis (*"[My suboxone doctor] knows and allows me to smoke weed, and we have a pretty good rapport"*) vs. providers who disallow it (*"They won't let me smoke weed which is my biggest complaint"*). A few other people referred to the policies and attitudes of treatment programs, again speaking preferentially about programs where cannabis use was permitted.

## Discussion

In this study examining the prevalence and content of cannabis-related posts in an opioid use subreddit and an opioid recovery subreddit, we found that cannabis mentions were more common in the recovery subreddit and that a primary motivation among those posting in the recovery subreddit about cannabis was the use of cannabis as a strategy to address opioid withdrawal symptoms. Although cannabis-related posts were relatively uncommon in these subreddits as a whole, cannabis-related posts were twice as prevalent in the recovery subreddit than in the opioid subreddit. Within these posts, posters disclosed several motivations for cannabis use in relation to opioid use. Frequent and unique phrases in the recovery subreddit alluded to use of cannabis as a treatment, substitution, or as one of several "comfort meds" for opioid withdrawal. In the opioid use subreddit, frequent and unique phrases alluded to use of cannabis in conjunction with opioids, with the most common motivations being polysubstance use, followed by withdrawal symptom management.

A primary finding of this study is that posters reported that the potential of cannabis to address OUD or reduce opioid use may derive from the use of cannabis to manage acute and persistent opioid withdrawal symptoms, a mechanism that has received less attention in the opioid-cannabis literature. While there are many interactions between the endocannabinoid and opioid systems, particularly with respect to rewarding properties of each drug [7, 42], cannabis itself does not address primary physiological opioid withdrawal mechanisms. The etiology of opioid withdrawal symptoms is related to complex adaptations in the opioid receptor-rich nervous and gastrointestinal systems resulting from repeated use of opioids [43]; withdrawal is thought to come from a sudden loss of tonic opioid receptor blockade resulting in

increased activation of norepinephrine and epinephrine. A more likely mechanism supporting our findings may be psychological relief, particularly anxiety relief and distracting or calming the mind from catastrophizing thoughts, especially if the person has had difficult withdrawal experiences before. For example, in a daily dairy study with chronic pain patients, negative affect and pain catastrophizing mediated the association between withdrawal symptoms and opioid craving [44]. As acute opioid withdrawal can be a distressing event, more research is needed on these cognitive and affective aspects of withdrawal, as well as non-opioid pharmacological and psychosocial options for withdrawal management.

In an online survey study of 200 adults who used cannabis and opioids and had experienced opioid withdrawal, nearly two-thirds reported using cannabis to treat withdrawal, reporting particular relief from anxiety, tremors, and trouble sleeping [45], which mirrors the accounts posted in the recovery subreddit. However, in an observational study of cannabis use among 116 patients undergoing methadone taper, opioid-withdrawal symptom scores did not differ significantly between patients who used cannabis and those who did not, even in time-lagged and sensitivity analyses [46]. While most self-reports in the present study focused on acute withdrawal symptoms, several posters reported using cannabis to manage what they described as longer-term PAWS symptoms [40, 41].

In contrast to substantial research interest in cannabis for pain management, pain was a rarely mentioned motivation for cannabis use in posts sampled from these subreddits for the current study. Posters also reported using cannabis at the same time as opioids to achieve a desired "high," though primarily in the non-treatment-seeking opioid subreddit. Posters also expressed uncertainty and ambivalence about the effectiveness and social acceptability of using cannabis while trying to abstain from opioids or in a methadone or buprenorphine treatment program. Many treatment program goals are focused primarily on achieving complete abstinence from all substances, and resistance to using pharmacotherapy for substance use disorder persists in some treatment modalities and 12-step programs [47–49]. Additionally, in the United States, federal Substance Abuse and Mental Health Services Administration (SAMHSA) guidelines require urine drug screens (including for THC) from patients in methadone and buprenorphine treatment programs at intake. Although many programs do not enact sanctions for positive urine drug screens for THC during treatment, some do, which may act as an additional barrier for engagement and retention in OUD treatment [50, 51]. Research from Canadian settings has found mixed evidence for the role of cannabis use in treatment retention [52, 53].

Previous research has found cannabis use motivations described by people who use illicit drugs include to "get high," relieve pain, and reduce opioid use [54], among a wide spectrum of therapeutic and recreational motivations that demonstrate unmet healthcare needs among opioid-using populations [55]. Despite the existence of FDA-approved medications for OUD including methadone, buprenorphine, and extended-release naltrexone, access to and engagement with these medications and other evidence-based psychosocial treatment modalities is limited [56–59]. Posters in the present study often reported using cannabis as one of several strategies to self-manage withdrawal symptoms in the context of limited interest in or access to OUD treatment.

There have been few randomized controlled trials on the use of cannabis or cannabinoids in OUD treatment [7]. In a double-blind, placebo-controlled study with 60 patients, the FDA-approved THC analog dronabinol was administered orally (10mg-30mg) as an adjunct to extended-release naltrexone for OUD during inpatient detoxification and for 5 weeks follow up, with findings of lower severity of opioid withdrawal symptoms in the dronabinol group compared to the placebo group, but similar rates of induction and treatment completion [60]. In a smaller human laboratory study with 12 opioid-dependent patients, higher doses of oral

dronabinol (20mg and 30mg vs. 5mg and 10mg) were associated with modest suppression of withdrawal severity but also with objectively measured increases in heart rate [61]. In a systematic review of 23 studies, McBrien et al. found no consensus that cannabis use was associated with reductions in opioid use or retention in treatment among patients in methadone maintenance therapy, though overall quality of evidence was low with a high risk of bias, in part due to primarily observational studies and large heterogeneity in cannabis use measurement [62]. A recent exploratory double-blind, placebo-controlled trial examined the effect of oral 400mg and 800mg CBD on drug cue-induced craving and anxiety among 42 abstinent patients with heroin use disorder, finding that participants in the CBD group had significantly reduced craving and anxiety compared with the placebo group [63]. In addition to the need for additional clinical and randomized controlled studies, more research is needed on the safety and efficacy of whole plant cannabis vs. specific cannabinoids (e.g., THC, THCA, CBD, CBN), dosages, and routes of administration (e.g., inhalation vs. ingestion), as well as participant groups from diverse OUD treatment and community settings (e.g., buprenorphine clinic patients, non-treatment seeking individuals).

There are several limitations to this research. A primary one is limited generalizability to people who use opioids, or who are in opioid use disorder treatment or in clinically or self-defined recovery, as we do not have demographic or geographic information about people who post on this pseudonymous forum. A related set of limitations is that posters may be more likely to share short term "success" stories with cannabis, which may in turn be due to positive expectancies or placebo effects. It is unknown how successful individuals were in the longer term or how often posters might report negative experiences with cannabis in these online venues. Another limitation is that our initial cannabis term filter list may have been incomplete (in not picking up specific slang or misspellings). Nevertheless, this exploratory analysis of naturalistic cannabis use in the context of opioid use has identified additional information about the potential impact of cannabis legalization on opioid use and related morbidity and mortality, especially in the context of unmet treatment needs for opioid use disorder.

Despite these limitations in generalizability, this examination of reports of naturalistic cannabis use in relation to opioid use identified withdrawal symptom management as a common motivation, as well as interest and uncertainty about cannabis among some people who use opioids or identify as in recovery from opioids. Future research is warranted with more structured survey and clinic-based assessments that examines the role of cannabis and cannabinoids in addressing both somatic and affective symptoms of opioid withdrawal.

## Acknowledgments

The authors wish to thank the members and moderators of these online communities for their open conversations.

## Author Contributions

**Conceptualization:** Meredith C. Meacham, Alicia L. Nobles, D. Andrew Tompkins, Johannes Thrul.

**Data curation:** Meredith C. Meacham.

**Formal analysis:** Meredith C. Meacham, Alicia L. Nobles, Johannes Thrul.

**Funding acquisition:** Meredith C. Meacham.

**Investigation:** Meredith C. Meacham.

**Methodology:** Meredith C. Meacham, Alicia L. Nobles, Johannes Thrul.

**Project administration:** Meredith C. Meacham.

**Writing – original draft:** Meredith C. Meacham.

**Writing – review & editing:** Meredith C. Meacham, Alicia L. Nobles, D. Andrew Tompkins, Johannes Thrul.

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
