## [Decision Letter · Decision Letter 0]

26 Oct 2021

PONE-D-21-16576“Is there anyone here who got clean with weed?”: Naturalistic cannabis use reported in online opioid and opioid recovery community discussion forumsPLOS ONE

Dear Dr. Meacham,

Thank you for submitting your manuscript to PLOS ONE. After careful consideration, we feel that it has merit but does not fully meet PLOS ONE’s publication criteria as it currently stands. Therefore, we invite you to submit a revised version of the manuscript that addresses the points raised during the review process.

We look forward to receiving your revised manuscript.

Kind regards,

Lucy J Troup, Ph.D

Academic Editor

PLOS ONE

Journal Requirements:

2. Thank you for stating the following in the Competing Interests/Financial Disclosure* (delete as necessary) section:

“I have read the journal's policy and the authors of this manuscript have the following competing interests: In the last 3 years, DAT has consulted for Alkermes, Inc. and Clinilabs and has served on a scientific advisory board for Alkermes, Inc.

The other authors have declared that no competing interests exist.”

We note that one or more of the authors are employed by a commercial company: Alkermes, Inc. and Clinilabs

Additional Editor Comments (if provided):

Thank you so much for your patience on this. COVID has impacted the manuscript review process significantly. I would like to concur with both reviewers that this is a nice addition to the existing literature and look forward to seeing a revised manuscript.

Reviewers' comments:

Reviewer's Responses to Questions

**Comments to the Author**

1. Is the manuscript technically sound, and do the data support the conclusions?

Reviewer #1: Yes

Reviewer #2: Yes

2. Has the statistical analysis been performed appropriately and rigorously? 

Reviewer #1: Yes

Reviewer #2: I Don't Know

3. Have the authors made all data underlying the findings in their manuscript fully available?

Reviewer #1: Yes

Reviewer #2: Yes

4. Is the manuscript presented in an intelligible fashion and written in standard English?

Reviewer #1: Yes

Reviewer #2: Yes

5. Review Comments to the Author

Reviewer #1: This manuscript is a really nice summary of social-media data on the co-use of opioids and cannabinoids, with data collected from people who are not trying to stop (in an opioid-use subreddit) and from people who are trying to stop (in an opioid-recovery subreddit).

My comments will make more sense if I de-anonymize myself: I'm David Epstein, who coauthored one of the papers that the authors cite in the Discussion. (Their summary, which is fine: "...in an observational study of cannabis use among 116 patients undergoing methadone taper, opioid-withdrawal symptom scores did not differ significantly between patients who used cannabis and those who did not, even in time-lagged and sensitivity analyses").

My stance in that paper (Epstein & Preston, 2015), based on my own group's outpatient data plus a literature review, was that whole-plant cannabis usually does little or nothing to alleviate opioid withdrawal. But my stance on most aspects of addiction, including this one, is that there's little to be gained by speaking in absolutes. So I welcome the social-media analyses in this manuscript and I think they're a useful update to the body of knowledge on that question. I also agree with the authors' interpretation, at least if I'm reading it correctly: of people posting to Reddit who have tried cannabis for opioid withdrawal, a substantial plurality say that it seems to provide some comfort. It's hard to know what’s expectation/placebo and what's not, but I'm perfectly ready to posit that some people are experiencing a real pharmacological benefit. This is the kind of heterogeneity I was pointing toward in my "Let's Agree to Agree" commentary in 2020, and it turns up in the human-laboratory literature reviewed by me and by the authors.

So, to get to the numbered/actionable part:

(1) I'd like to see the issue (does cannabis truly relieve symptoms of opioid withdrawal?) addressed a little more forcefully in the Discussion, with some attempt at a takeaway message like the one I tried to give in my last long paragraph here. In formulating that message, it might be useful to consider CBD separately from whole-plant cannabis, just because the two things lead to such different subjective experiences. Some of the subreddit posts make a point of saying that unwanted effects from whole-plant cannabis can become problematic.

(2) This comment is numbered, but possibly not actionable. My main reservation about the manuscript is that it's mostly a travelogue of the space covered by the two subreddits--it's reasonably organized, but it doesn't set up a series of urgent/compelling questions and then give strong answers. Not having read all the downloaded material from the subreddits, I don't know whether it's possible to move the presentation in that direction. If there are elevator-summary points to be pulled from the material, I'd like to see them emphasized. If the material doesn't support that approach, I'm fine with the inherent value of what's here.

Reviewer #2: 1. A fascinating and well done study using a source of data rarely examined.

2. Authors reference a paper that describes legalization of marijuana correlated with decreased opiate overdose deaths. Please add the follow- up paper that extended dates of study and showed this correlation no longer held.

3. Authors may want to answer the question of the paper title. It appears to be that marijuana may help with detox symptoms but there is almost no support for the issue of whether marijuana helps to prevent relapse to opiates. I would think that if participants in the opiate recovery group found marijuana to be helpful as a harm reduction strategy to prevent return to opiate use, they would present rave reviews of this on their Reddit posts. Thus the absence of any positive comments may be the most significant finding of this study. Similar to my report of 26 adolescent/young adult patients with OUD where smoking marijuana precipitated a relapse to opiate use, one poster described that smoking marijuana triggered his relapse to opiates.

4. This study adds to the literature that marijuana containing THC is not the answer to our opiate epidemic.

6. PLOS authors have the option to publish the peer review history of their article (what does this mean?). If published, this will include your full peer review and any attached files.

Reviewer #1: **Yes: **David H. Epstein

Reviewer #2: **Yes: **Steven L. Jaffe, MD Professor Emeritus of Psychiatry, Emory University School of Medicine and Clinical Professor of Psychiatry, MoreHouse School of Medicine

MD

---

## [Author Response · Author response to Decision Letter 0]

16 Dec 2021

Dear Dr. Troup and Reviewers:

Thank you for the opportunity revise and resubmit our manuscript with an updated title of “‘I got a bunch of weed to help me through the withdrawals’: Naturalistic cannabis use reported in online opioid and opioid recovery community discussion forums” for consideration in PLOS ONE. 

We appreciate the comments from the reviewers and our responses to changes to the manuscript are noted below in bold. In particular, we endeavored to highlight the takeaway message that online posters perceived cannabis to be helpful addressing affective withdrawal symptoms and that cannabis use was one of many strategies to self-manage withdrawal.

We look forward to hearing from you and thank you for your efforts reviewing our manuscript.

Reviewer #1: This manuscript is a really nice summary of social-media data on the co-use of opioids and cannabinoids, with data collected from people who are not trying to stop (in an opioid-use subreddit) and from people who are trying to stop (in an opioid-recovery subreddit).

My comments will make more sense if I de-anonymize myself: I'm David Epstein, who coauthored one of the papers that the authors cite in the Discussion. (Their summary, which is fine: "...in an observational study of cannabis use among 116 patients undergoing methadone taper, opioid-withdrawal symptom scores did not differ significantly between patients who used cannabis and those who did not, even in time-lagged and sensitivity analyses").

My stance in that paper (Epstein & Preston, 2015), based on my own group's outpatient data plus a literature review, was that whole-plant cannabis usually does little or nothing to alleviate opioid withdrawal. But my stance on most aspects of addiction, including this one, is that there's little to be gained by speaking in absolutes. So I welcome the social-media analyses in this manuscript and I think they're a useful update to the body of knowledge on that question. I also agree with the authors' interpretation, at least if I'm reading it correctly: of people posting to Reddit who have tried cannabis for opioid withdrawal, a substantial plurality say that it seems to provide some comfort. It's hard to know what’s expectation/placebo and what's not, but I'm perfectly ready to posit that some people are experiencing a real pharmacological benefit. This is the kind of heterogeneity I was pointing toward in my "Let's Agree to Agree" commentary in 2020, and it turns up in the human-laboratory literature reviewed by me and by the authors.

R: Thank you for your kind summary of our manuscript and the reference to your commentary. We appreciate the attention to closing questions in the commentary of “when, for whom, and to what extent.” In conducting this study, it is those questions we hoped to begin to address by drawing from these first person reports in this online setting. 

So, to get to the numbered/actionable part:

(1) I'd like to see the issue (does cannabis truly relieve symptoms of opioid withdrawal?) addressed a little more forcefully in the Discussion, with some attempt at a takeaway message like the one I tried to give in my last long paragraph here. In formulating that message, it might be useful to consider CBD separately from whole-plant cannabis, just because the two things lead to such different subjective experiences. Some of the subreddit posts make a point of saying that unwanted effects from whole-plant cannabis can become problematic.

R: We have revised and reorganized the opening of the discussion section (page 19) to highlight the perceived helpfulness in cannabis addressing affective withdrawal symptoms and that cannabis use was one of many strategies to self-manage withdrawal. 

New opening (lines 392-396): “In this study examining the prevalence and content of cannabis-related posts in an opioid use subreddit and an opioid recovery subreddit, we found that cannabis mentions were more common in the recovery subreddit and that a primary motivation among those posting in the recovery subreddit about cannabis was the use of cannabis as a strategy to address opioid withdrawal symptoms.”

In the second paragraph (lines 403-418), we expand on the potential physiological mechanisms for this finding, adding two citations on opioid withdrawal etiology and negative affect in withdrawal: 

Kosten TR, Baxter LE. Review article: Effective management of opioid withdrawal symptoms: A gateway to opioid dependence treatment. Am J Addict. 2019;28(2):55-62.

Bruneau A, Frimerman L, Verner M, Sirois A, Fournier C, Scott K, et al. Day-to-day opioid withdrawal symptoms, psychological distress, and opioid craving in patients with chronic pain prescribed opioid therapy. Drug and alcohol dependence. 2021;225:108787.

We also added to the results section (in Table 4) an evaluation of whether the poster seemed to find cannabis helpful and a summary of motivations (Figure 1). Regarding CBD, we reviewed these CBD-mentioning posts to see if CBD was mentioned alone or with THC-containing cannabis and reported this in the text. We also noted in the future research sections the importance of examining whole plant vs. specific cannabinoids. 

(2) This comment is numbered, but possibly not actionable. My main reservation about the manuscript is that it's mostly a travelogue of the space covered by the two subreddits--it's reasonably organized, but it doesn't set up a series of urgent/compelling questions and then give strong answers. Not having read all the downloaded material from the subreddits, I don't know whether it's possible to move the presentation in that direction. If there are elevator-summary points to be pulled from the material, I'd like to see them emphasized. If the material doesn't support that approach, I'm fine with the inherent value of what's here.

R: This analysis is indeed largely exploratory but in the revisions we have endeavored to highlight the reported use of cannabis to self-manage affective opioid withdrawal symptoms in the context of unmet treatment needs. 

Reviewer #2: 1. A fascinating and well done study using a source of data rarely examined.

R: Thank you.

2. Authors reference a paper that describes legalization of marijuana correlated with decreased opiate overdose deaths. Please add the follow- up paper that extended dates of study and showed this correlation no longer held.

R: We have added that 2019 paper and briefly noted its findings. We also added a reference to a 2018 systematic review of ecological and epidemiologic studies examining this relationship. 

Campbell, G., Hall, W., & Nielsen, S. (2018). What does the ecological and epidemiological evidence indicate about the potential for cannabinoids to reduce opioid use and harms? A comprehensive review. International review of psychiatry (Abingdon, England), 30(5), 91–106. https://doi.org/10.1080/09540261.2018.1509842

3. Authors may want to answer the question of the paper title. It appears to be that marijuana may help with detox symptoms but there is almost no support for the issue of whether marijuana helps to prevent relapse to opiates. I would think that if participants in the opiate recovery group found marijuana to be helpful as a harm reduction strategy to prevent return to opiate use, they would present rave reviews of this on their Reddit posts. Thus the absence of any positive comments may be the most significant finding of this study. Similar to my report of 26 adolescent/young adult patients with OUD where smoking marijuana precipitated a relapse to opiate use, one poster described that smoking marijuana triggered his relapse to opiates.

R: We have revised the quotation in paper title to emphasize the finding regarding withdrawal symptom management as a motivation for cannabis use “I got a bunch of weed to help me through the withdrawals”, as the data in this study are not sufficient to address that question directly. 

Regarding the presence or absence of positive comments, of the people who disclosed using cannabis to manage withdrawal symptoms, nearly all of had positive attitudes towards it. We note that the perceived effectiveness of cannabis alone was often difficult to determine, as people often listed cannabis as one of several strategies. 

To clarify this, we recoded the posts labeled as relevant to withdrawal and pain whether the poster found cannabis to be helpful on its own, with other strategies, unclear, or not helpful, with summaries in Table 4. We also added these passages:

Line 262: “Most people, but not everyone, who described using cannabis for withdrawal symptom management described it as being helpful, but often in the context of a list of medications and other strategies.”

Line 334: “While the majority of posts describing cannabis use for withdrawal symptom management were positive towards cannabis (42/55), it was not always clear whether the helpfulness was due to cannabis alone or to other strategies.”

Line 374: “As with withdrawal symptom management, cannabis was one several strategies people used to manage pain.”

4. This study adds to the literature that marijuana containing THC is not the answer to our opiate epidemic.

R: While we agree that THC is not the answer to the opioid epidemic, we hope this study adds to literature on how people are managing opioid withdrawal outside of medical settings as well as the general interest and uncertainty about cannabis among people who use opioids.

---

## [Decision Letter · Decision Letter 1]

24 Jan 2022

“I got a bunch of weed to help me through the withdrawals": Naturalistic cannabis use reported in online opioid and opioid recovery community discussion forums

PONE-D-21-16576R1

Dear Dr. Meacham,

We’re pleased to inform you that your manuscript has been judged scientifically suitable for publication and will be formally accepted for publication once it meets all outstanding technical requirements.

Kind regards,

Lucy J Troup, Ph.D

Academic Editor

PLOS ONE

Additional Editor Comments (optional):

Reviewers' comments:

Reviewer's Responses to Questions

**Comments to the Author**

1. If the authors have adequately addressed your comments raised in a previous round of review and you feel that this manuscript is now acceptable for publication, you may indicate that here to bypass the “Comments to the Author” section, enter your conflict of interest statement in the “Confidential to Editor” section, and submit your "Accept" recommendation.

Reviewer #1: All comments have been addressed

Reviewer #2: All comments have been addressed

2. Is the manuscript technically sound, and do the data support the conclusions?

Reviewer #1: Yes

Reviewer #2: (No Response)

3. Has the statistical analysis been performed appropriately and rigorously? 

Reviewer #1: Yes

Reviewer #2: (No Response)

4. Have the authors made all data underlying the findings in their manuscript fully available?

Reviewer #1: Yes

Reviewer #2: (No Response)

5. Is the manuscript presented in an intelligible fashion and written in standard English?

Reviewer #1: Yes

Reviewer #2: (No Response)

6. Review Comments to the Author

Reviewer #1: I'm satisfied with the authors' responses and revisions, and I like the new Venn diagram. Nice work!

Reviewer #2: (No Response)

7. PLOS authors have the option to publish the peer review history of their article (what does this mean?). If published, this will include your full peer review and any attached files.

Reviewer #1: **Yes: **David H. Epstein

Reviewer #2: No

---

## [Editor Report · Acceptance letter]

31 Jan 2022

PONE-D-21-16576R1 

*“I got a bunch of weed to help me through the withdrawals”:* Naturalistic cannabis use reported in online opioid and opioid recovery community discussion forums 

Dear Dr. Meacham:

I'm pleased to inform you that your manuscript has been deemed suitable for publication in PLOS ONE. Congratulations! Your manuscript is now with our production department. 

Kind regards, 

on behalf of

Dr. Lucy J Troup 

Academic Editor

PLOS ONE